

# Unifying biological field observations to detect and compare ocean acidification impacts across marine species and ecosystems: What to monitor and why

Steve Widdicombe[1], Kirsten Isensee[2], Yuri Artioli[1], Juan Diego Gaitán-Espitia[3], Claudine Hauri[4], Janet A. Newton[5], Mark Wells[6,7], Sam Dupont[8,9]

[1]Plymouth Marine Laboratory (PML), Plymouth, PL1 3DH, United Kingdom
[2]Intergovernmental Oceanographic Commission of the United Nations Educational, Scientific and Cultural Organization, Paris, 75732, France
[3]The Swire Institute of Marine Science, School of Biological Sciences, The Hong Kong University, Hong Kong, China
[4]International Arctic Research Center, University of Alaska Fairbanks, Fairbanks, AK 99775-0100, USA
[5]Applied Physics Laboratory and College of the Environment, University of Washington, Seattle, WA 98105-6698, USA
[6]School of Marine Sciences, The University of Maine, Orono, ME 04469-5706, USA
[7]State Key Laboratory of Satellite Ocean Environment Dynamics, Second Institute of Oceanography, Ministry of Natural Resources, Hangzhou, China
[8]Department of Biological and Environmental Sciences, University of Gothenburg, Fiskebäckskil, 45178, Sweden
[9]Radioecology Laboratory International Atomic Energy Agency (IAEA), Marine Laboratories, 98000, Principality of Monaco

*Correspondence to*: Sam Dupont (sam.dupont@bioenv.gu.se)

**Abstract.** Approximately one quarter of the $CO_2$ emitted to the atmosphere annually from human activities is absorbed by the ocean, resulting in a reduction of seawater pH and shifts in seawater carbonate chemistry. This multi-decadal process, termed "anthropogenic ocean acidification" (OA) has been shown to have detrimental impacts on marine ecosystems. Recent years have seen a globally coordinated effort to measure the changes in seawater chemistry caused by OA, with best practices now available for these measurements. In contrast to these substantial advances in observing physico-chemical changes due to OA, quantifying their biological consequences remains challenging, especially from in-situ observations under real-world conditions. Results from two decades of controlled laboratory experiments on OA have given insight into the likely processes and mechanisms by which elevated $CO_2$ levels affect biological process, but the manifestation of these process across a plethora of natural situations has yet to be explored fully. This challenge requires us to identify a set of fundamental biological and ecological indicators that are i) relevant across all marine ecosystems, ii) have a strongly demonstrated link to OA, and iii) have implications for ocean health and the provision of ecosystem services with impacts on local marine management strategies and economies. This paper draws on the understanding of biological impacts provided by the wealth of previous experiments, as well as the findings of recent meta-analyses, to propose five broad classes of biological indicators that, when coupled with environmental observations, including carbonate chemistry, would allow the rate and severity of biological change in response to OA to be observed and compared. These broad indicators are applicable to different ecological systems, and the methods for data analysis suggested here would allow researchers to combine biological response data across regional and global scales by correlating rates of biological change with the rate of change in carbonate chemistry parameters. Moreover, a method using laboratory observation to design an optimal observing strategy (frequency and duration) and observe meaningful biological



rates of change highlights the factors that need to be considered when applying our proposed observation strategy. This innovative observing methodology allows inclusion of a wide diversity of marine ecosystems in regional and global assessments and has the potential to increase the contribution of OA observations from countries with developing OA science capacity.

**1. Introduction**

Anthropogenic perturbations to the earth system directly impact the health of the world's ocean, its biochemical dynamics, its ecological properties and natural resources, and consequently the provision of ecosystem services. Ocean change includes so-called 'slow onset events', such as ocean warming, acidification, deoxygenation, sea level rise, and glacial retreat, as well as an increase of extreme weather and biogeochemical anomalous conditions, such as marine heat waves (Hobday et al., 2016),

storms, and extreme acidification and deoxygenation events (Gruber et al., 2021).

Today, the ocean absorbs about one quarter of the $CO_2$ emitted to the atmosphere annually by human activities, resulting in a reduction of seawater pH (Friedlingstein et al., 2019, 2020). This multi-decadal process, termed "anthropogenic ocean acidification", causes additional chemical changes, including altering the speciation of dissolved inorganic carbon (IPCC 2019). Other regional and local changes also affect carbonate chemistry, such as enhanced upwelling, sea ice melt, riverine

inputs, or increased temperature. Marine species and ecosystems respond to changes in carbonate chemistry, regardless of the driving mechanism, so we use the term ocean acidification (OA) here more broadly to refer to all acidification process be they anthropogenically-derived global, regional or local processes.

Although there is an accepted recognition of OA as a threat to ocean health and its detrimental and non-linear impacts on marine species and ecosystems, the initial focus of international networks, projects and organizations was to adequately

describe, quantify, and understand the chemical changes associated with OA (Newton et al., 2015). Substantial progress in this direction has been made (Tilbrook et al., 2019), notwithstanding that many geographical regions still lack sufficient capability, information and data to determine local OA conditions. Best practices and standard operating mechanisms are now available to the global scientific community to assess spatial and temporal patterns in pH and carbonate chemistry, to quantify trends and rates of change, and to identify areas with strong signals of OA chemical impacts (IOC-UNESCO 2019, and references

therein).

The second, even more challenging goal for the scientific community is to identify the impacts of OA on marine life, as they occur in real world situations, over timescales that are appropriately matched to the rate at which the chemical changes are occurring and over spatial scales that are appropriate for the biological processes being impacted. Currently less than 10% of the published literature addressing the impacts of OA on species and ecosystems have included biological field observations

in their studies, with the majority of scientific publications that have focused on field observations are in areas of unusually high $CO_2$ levels or laboratory experiments (OA-ICC source, February 2022). Scaling up the knowledge gained from two decades of experimental results, from both laboratories and from natural high $CO_2$ environments, such as $CO_2$ seeps and





upwelling regions, to appreciate the chronic impacts of OA across a variety of marine ecosystems will inevitably require a significant increase in in-situ biological monitoring that is specifically relevant to OA.

Marine environmental in-situ observations are a prerequisite for ecosystem-based management and is essential to ensure that sustainable management targets are met. Traditionally, monitoring and observations provide the context to marine science and have allowed development of a critical scientific understanding of the marine environment and the impacts that humans are having on it (Bean et al., 2017). Biological observations are also an important complement to experimental studies as it sets the boundary conditions within which to measure the rate of change (speed at which a given process changes over a specific

period of time) in any given biological parameter (e.g. biodiversity; see Luypart et al., 2020). Whilst traditional biological observations programs focus on identifying and quantifying (counting and weighing) a wide range of taxa and communities (Muller-Karger et al., 2018), such programs are rarely linked to direct OA field observations. This means that presently, applied biological observation strategies have difficulties in attributing the change, or a proportion of it, to OA as there is little specific focus on those biological parameters most affected by carbonate chemistry, as well as a lack of supporting carbonate chemistry

data. Long-term studies that appropriately consider the biological changes driven by OA are still missing, which limits the development of appropriate marine management strategies to either mitigate stress or adapt to OA impacts.

The biggest obstacle to increasing the overall effectiveness of biological observations for OA is currently a lack of clear guidance on what biological parameters to measure, and importantly, how such observations can be compared with others to generate greater understanding of the trends, hotspots and patterns occurring over local, regional and global scales. To address

this challenge, we need to identify a clear subset of biological variables and indicators that provide quantitative information across all ecosystems, as well as a method to, not only understanding the rate and severity of the local impact, but also compare multiple datasets to deliver a greater holistic understanding of OA's biological impacts, regionally and globally. Such methods will rely on biological observations being coupled with a suite of environmental observations including, but not limited to, measures of carbonate chemistry, into integrated observing activities.

Here, we present an innovative approach that stems from considering five fundamental ecosystem traits that span across marine ecosystems and which have been identified as potentially sensitive to OA by a significant body of previously published studies. Firstly, we will introduce our approach and describe why such a trait-based approach is appropriate, Secondly, we will illustrate why these five specific traits have been identified and provide examples of what parameters and processes can be measured in each of the traits (e.g. Kroeker et al., 2013). Thirdly, we will present an example of how biological observing data can be

coupled with environmental data to compare the results from multiple studies or observations, in order to better understand the biological impacts of OA, identify trends and hotspots, appreciate the generality of response and identify the extent to which OA is driving the changes observed in relation to other environmental stresses. We recognize there could be several different methods that would be appropriate for such data analyses and encourage researcher to employ those methods most suitable to address their specific research question. Finally, we will use a combination of conceptual and real data examples to illustrate

the key factors and assumptions that need to be considered when applying the monitoring approaches proposed here.



## 2. Biological field observations using traits as indicators of OA

Quantifying rates of change in carbonate chemistry due to anthropogenic inputs at a suite of globally distributed locations is
one of the main goals of the observing community (Newton et al., 2015) and is included in the indicator for the Sustainable
Development Goal target 14.3. In principle, using a similar approach to evaluate biological sensitivity to OA would allow
direct comparability between the growing availability of carbonate chemistry data and biological observations. However, while
the measurement of carbonate chemistry is largely limited to a small number of parameters (pH, DIC, $p$CO$_2$, total alkalinity),
all of which can be calculated with different degrees of uncertainty from direct measurements of any two of these parameters,
measurement of biological impacts includes a much wider range of parameters, each with different scales and units. Thus,
while standardizing the observation of carbonate chemistry changes is relatively straightforward, standardizing the
measurement of biological impacts in a way that would allow broad scale comparisons between regions and ecosystems is
considerably more challenging.

The first challenge for delivering a truly global assessment of OA impacts is to identify a set of fundamental biological
parameters that a) are not linked to any given model organism; b) are relevant across different biological systems; c) have a
strongly demonstrated link to OA; and, d) have implications for the overall function of ecosystems and the provision of
ecosystem services. OA alters multiple properties of the seawater carbonate system, including pH, speciation of DIC, and
carbonate saturation state, that individually, or in combination, affect marine organisms. Consideration of only one of these
parameters then may obscure the mechanistic origin of any observed OA-driven change in biological attributes. While other
drivers, such as temperature, nutrients, oxygen, etc., may exert proximate control over changes in many marine ecosystems, it
is important to recognize that OA may have a major role in modulating these consequences in most marine waters, while being
the dominant driver for ecosystem change in others.

In addition to the influence of other climate drivers, it should be noted that biological impacts can be heavily moderated by
other environmental and ecological factors such as energy supply and other drivers and stressors such as nutrient and food
content. In regions experiencing multiple elevated environmental stressors (e.g., increased sedimentation, organic or inorganic
contaminants) it will be more difficult to disentangle the longer-term OA effects from those potential confounding effects.
However, this does not negate the importance of collecting OA impact data from these areas, but it does emphasize the
importance of collecting additional relevant environmental data to aid interpretation of the observed biological responses. In
addition, for indicators measured at the species level, different life stages may also show different responses (e.g. Kroeker et
al., 2013; Bednarsek et al., 2019; 2021a,b). These measures for adults can give regional insights to how ecosystems have
responded to OA pressures, while measuring these aspects on juvenile stages may help to presage the future changes in these
ecosystems. So again, the importance of collecting environmental, ecological or physiological contextual data cannot be over
emphasized.



**3. Observing five traits that are influenced by OA**

Based on previously published meta-analysis and reviews (e.g. Doney et al., 2020; Figuerola et al., 2012; Hancock et al., 2020; Kroeker et al., 2013; Turley and Gattuso, 2012), we delineate five fundamental ecosystem traits and their suite of observable indicators (Fig. 1): 1) calcified organisms and calcification, 2) autotrophs and primary production, 3) heterotrophs and secondary production, 4) biodiversity and community structure, and 5) genetic adaptation. The specific choice of indicator at different sites will depend on the questions or concerns of the investigator, but should relate to fitness of species, the health of marine populations and communities, and if relevant, associated with key ecosystem services. The feasibility of obtaining these observations over the long term also must be considered. Detection of OA-induced change in these indicators hinges upon the variability of the data, so a first step is to apply recognized best practices for biological measurements (Katsanevakis et al., 2012; Bean et al., 2017). Even so, natural "noise" of the system, in terms of short-term environmental changes (weather) and organismal plasticity, must also be taken into account when selecting indicators because it strongly influences the time of emergence of long-term trends.

The specific biological observations listed in this section are done so as examples of potential parameters to measure and are not exhaustive. They provide a suitable framework on which to build both intensive and pragmatic field observation programs that are best suited for local capacities and funding. These methodological inventories undoubtedly will evolve as future technologies emerge and become more widely available over the coming decades (Bean et al., 2017), but the central ecosystem traits they manifest, combined with high quality carbonate system data, will enable quantification of OA-driven changes in marine ecosystems. The goal is to determine the relationship between rates of chemical and biological changes, enabling comparisons of those relationships across diverse and geographically-distributed ecosystems.

**3.1. Calcified Organisms and Calcification**

Rationale: The use of calcium carbonate ($CaCO_3$) structures for structural support and protection is widespread amongst marine organisms (e.g., Monteiro et al., 2016; Fitzer et al., 2019; Clark et al., 2020). As changes in seawater chemistry have been demonstrated to affect calcium carbonate structures and biomineralization rates (see refs in Fitzer et al., 2019), analysis of effects from acidification on calcifiers is perhaps the most mature of OA biological impact studies (e.g., Hofmann et al.; 2008, Vézina and Hoegh-Guldberg, 2008; Wittmann and Pörtner; 2013, Bednaršek et al., 2014, 2017, 2021a; Osborne et al., 2020). OA can affect the ratio of calcifiers to non-calcifiers in an ecosystem, aspects of calcified structures, and measured rates involving calcium carbonate, including both the deposition and dissolution of calcium carbonate (Fitzer et al., 2019).

Indicator categories and observations:

● Relative prevalence and success of calcifying organisms within an ecosystem: Changes in biomass/abundance/percent cover of biocalcifying species, compared to non-calcifying species, measured across either fine scale (meters) or broad scales (kilometres) can be used as a primary indicator of OA stress, once other known factors are ruled out (e.g., eutrophication).



Inorganic to organic biomass ratios (PIC:POC) of individual organisms, populations or whole assemblages can also provide a sensitive measure of relative changes in the presence of biocalcifying organisms within an ecosystem (Thomsen et al., 2010).

● Calcified Biostructure Morphology: The state, structure and mineral composition of calcified biostructures yields
information about the ability of organisms to maintain the balance of calcification over dissolution, and thus is an indicator of biological stress. Calcified structure morphology or function can be assessed through measures such as weight, density, damage or abnormality, dissolution severity, or strength (e.g., Bednarsek et al., 2017; 2021a; Osborne et al., 2020).

● Rates of Calcification: Calcifying conditions in benthic habitats can undergo large shifts over short (hourly, diurnal) time scales related to physical forcing (e.g., tides, light levels). Measured rates of calcification or dissolution on organismal to
ecosystem scales help to discern short-term (semi-instantaneous) responses of biocalcifying organisms to present conditions (e.g., Goreau, 1959; Cohen et al., 2017; Schoepf et al., 2017). These can be accomplished at larger scales with water-mass tracking of observed changes in TA and DIC (along with T, S) to estimate net community calcification (Fitzer et al., 2019), but also at organismal scales with in-situ approaches (Melzner et al., 2010; Dellisanti et al., 2020).

**3.2. Autotrophs and Primary Production**

Rationale: Primary production, in all of its myriad forms, serves as the foundation for all marine ecosystems. Quantifying its susceptibility to OA is crucial because primary production ultimately provides the energy for all trophic levels to adjust to OA-induced stress. Increasing $CO_2$ can modulate primary production in confounding ways. OA shifts the carbonate system to yield higher $pCO_2(aq)$ which is a fundamental fuel for photosynthesis that often becomes limiting in high productivity systems
(Beardall et al., 2009). Many primary producers (but not all) initiate metabolically-costly carbon concentrating mechanisms that divert energy from growth and reproduction. So, increases in $pCO_2(aq)$ then have the potential to stimulate carbon fixation rates in pelagic and benthic primary producers (Raven et al., 2014), including in marine animal-algal symbioses (Dupont et al., 2012). However, decreasing external and cytosolic pH can disrupt a broad suite of physiological and biochemical factors, including proton pumps, cellular membrane potential, enzyme activity, and energy partitioning (Beardall and Raven, 2004;
Giordano et al., 2005). OA also may influence the chemical availability of essential micronutrients (e.g., trace metals, Gledhill et al. 2015), thereby potentially affecting the nutritional support of primary production in some coastal and offshore waters (Hutchins and Boyd, 2016). Another mode of potential impact is that transmembrane proton gradients regulate flagellar motion and thus cell motility (Manson et al., 1977), so OA may might reduce the access of vertically migrating flagellated phylogenetic groups to nutrients, while simultaneously increasing their susceptibility to predation (Hallegraeff et al., 2012). Despite the
likely complexity of potential OA effects on autotrophs and primary production, the responses will be underlain by three chief indicators.

Indicator categories and observations:

● Biomass/Standing Stock: The total biomass of primary producers in pelagic and benthic environments often are used to signal the energy available for transfer to higher trophic levels. Although methods for quantifying biomass or standing stock





vary depending on the environment or organism of interest, they can include measures of total chlorophyll a (chl a) concentrations, phytoplankton cell abundance, microphytobenthos biomass (e.g., chl a per area of surficial sediments), as well as the biomass of macroalgae and seagrasses. A central benefit of this metric is that it is less susceptible to interannual variability, and thus gives a more time-integrated insight into the status of communities and regions.

    ●     Productivity: Lower levels of biomass at times can mask high carbon turnover rates, so measuring carbon fixation

rates or planktonic, macroalgal or seagrass growth rates can provide more sensitive and comprehensive insight to energy flows in the system. In some cases, whole community production rates can be estimated by measuring net carbon and oxygen dynamics. However, such estimates are prone to short-term variability at hourly, weekly, and seasonal intervals. Establishing trends associated with OA will require longer time series.

    ●     Phenology: Any change in the rates of primary production will alter nutrient dynamics and so it will be important to

measure the timing of blooms or other rapid growth periods. If these shifts in phenology become significant, they can lead to temporal disconnect between the timing of high primary producer biomass and sensitive (e.g., larval) life cycle stages of secondary producers that depend upon this energy source.

### 3.3. Heterotrophs and Secondary Production

Rationale: There is good evidence that long-term exposure to elevated $p$CO$_2$ levels increases metabolic energy demand in many marine organisms (e.g., Stumpp et al., 2011; Pan et al., 2015; Jager et al., 2016). This additional energy is needed to support increased acid-base balance activities (e.g. increased cellular proton pump activity to counter acidosis of intracellular fluids) (e.g. Stumpp et al., 2012); increased calcification to counter increased calcium carbonate dissolution (e.g. Ventura et al., 2016), and increased physical activities (e.g. increased burrow ventilation) (Donohue et al., 2012). These additional energy

expenditures leave less energy available to invest in other key processes, including protein synthesis and growth. Conversely, any increase in energy demands may be alleviated in part or wholly by increased food supply from CO$_2$-enhanced primary production, leading to overall increases in heterotroph populations and productivity and thereby changes in ecosystem function. For humans, reductions in marine secondary production may have profound implications for fisheries and aquaculture.

Indicator categories and observations:

●     Biomass/Standing Stock: The total biomass or standing stock of secondary producers in pelagic and benthic environments is not only a sentinel of the available energy transfer upwards into food webs, but also serves as a top-down integrated measure of effective primary production; i.e., that autotrophic growth utilized by grazing versus that produced by noxious, toxic, or unpalatable autotrophs. The total heterotrophic community abundance, ideally quantified as biomass per individual but also as numbers of individuals, average body sizes, or percent cover (e.g., sponges, colonial organisms), should

be quantified to estimate total heterotrophic biomass on appropriate seasonal or annual time scales. Splitting the quantification

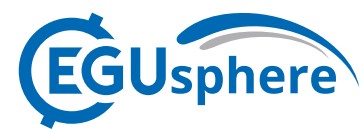

of abundance and biomass into major functional or species groups can provide greater detail and resolution for identifying those aspects of the community or ecosystem that are most affected.

● Productivity: As with primary producers, the observed biomass or standing stock of secondary producers can be the consequence of rapid or more sluggish growth/predation cycling, so quantifying the rates of secondary production at either community or species scales provides important additional insights to change within environments. There exists a wide range of methodologies to calculate gross estimates of pelagic and benthic secondary production from in-situ techniques to algorithms (Brey, 2012). It may be useful to emphasize the observation of selected heterotrophic taxa that play a strong structuring role within the local ecosystem, or that have high cultural or commercial value. Through its integrative nature, the estimates of secondary productivity are likely to exhibit less temporal variability than that of primary productivity, although the metrics of biomass or standing stock still are the preferable indicator for establishing OA induced effects on heterotrophs and secondary production.

● Phenology: Most marine secondary producers have planktonic life stages that are coupled to environmental cues which signal food abundance (e.g., temperature, light), but this pairing may fail if OA-driven shifts in primary production become substantial. The same may be true for other segments of the food web (e.g., migration patterns of higher organisms). As with primary producers then, quantifying any changes in the phenology of secondary producers will be important for assessing the potential influence of OA on marine ecosystems (Calbet et al., 2014).

### 3.4. Biodiversity and community structure

Rationale: Many of the processes mentioned in the previous paragraphs result in changes in biodiversity and community structure. There is compelling evidence that the sensitivity of marine organisms to all aspects of OA varies enormously among species (Vargas et al., 2017, 2022). This variability is created via the combination of different physiological and ecological traits exhibited by the different taxa. Such interspecific variability could allow OA to act directly as a strong selective pressure, decreasing biodiversity directly through species loss or indirectly through a host of processes, such as disrupting competition (e.g. space occupation or resource allocation) or trophic interactions (e.g. predation and grazing), decreasing energy generation and flow, degrading critical biogenic habitats, or increasing susceptibility to pathogens and diseases (Widdicombe and Spicer, 2008; Sunday et al., 2017; Bibby et al., 2008). Evidence from the paleontological record supports this assumption, with large species extinctions and biodiversity losses being associated with periods of extreme climate change events (Twichett, 2007).

Indicator Categories and observations:

● Taxonomic diversity and community composition: This represents the manifestation of all the direct and indirect impacts acting upon all of the individuals within a community. It is an established metric used in many existing marine monitoring programmes and impact assessments. The essential observations to be made are to identify and quantify (number or biomass) the species present within a community or assemblage at any given time. From these species abundance/biomass



data, it is possible to calculate a wide variety of indices, each focusing on specific aspects of community structure and
biodiversity (e.g. community abundance/biomass, species richness, evenness, taxonomic relatedness, Gaston and Spicer,
2004). This approach is applicable to all types of communities, including microbial assemblages. Whilst identification of
individuals to the species level is desirable this is not always possible, however, identification at lower taxonomic resolution
still allows the relative effects of OA to be assessed on specific taxonomic or functional groups.

● Functional or trait diversity: Similar to taxonomic biodiversity metrics described in the previous section, metrics of
functional or trait diversity again require the collection of species abundance/biomass data as detailed above. However, in this
instance individual organisms are grouped, not by their taxonomic relationships, but by their shared functional, ecological or
behavioural traits. The same biodiversity metrics as used for describing taxonomic diversity can then also be applied to these
aggregated data to generate estimates of functional or trait diversity. The benefit of this approach is that it provides information
on the potential function and performance of ecosystems and is particularly employed in the study of microbial biodiversity
(Krause et al., 2014).

## 3.5. Genetic diversity

Rationale: Understanding the impact of OA on marine biodiversity requires not only characterizing how these changes will
affect ecosystems, but also how populations will respond via acclimation and adaptive evolution (Sunday et al., 2014). OA is
an important driver of phenotypic (e.g., morphology, physiology, life-history and even behavior) (Kroeker et al., 2013) and
genetic change (functional and structural) (e.g., de Wit et al., 2016; Lloyd et al., 2016). OA is therefore an important component
of natural selection. The selective effects of OA can influence micro-evolutionary dynamics by altering demographic
parameters (Bramanti et al., 2015), genetic diversity (Lloyd et al., 2016), and the molecular regulation of functional pathways
(de Wit et al., 2016; Runcie et al., 2016) in marine organisms. Every species and population has an adaptation potential to OA
that is mostly dependent on genetic diversity (Sunday et al., 2014). Characterizing the present level and rate of changes in
relevant aspects of genetic diversity is key to understanding the future of marine ecosystems. While this signature can be
specific to OA, the signal can vary in strength and direction across temporal and spatial scales depending on the particular
genetic and environmental backgrounds of natural populations (Pespeni et al., 2013; Calosi et al., 2017; Gaitan-Espitia et al.,
2017).
Progress in high-throughput sequencing technologies allow rapid, cost-effective and informative measurements of molecular
genetic variation (functional and neutral) in natural populations. These techniques have the potential to track, assess and
disentangle the specific OA signature at the molecular level and estimate the rate of micro-evolution in response to OA across
population and species.

Indicator categories and observations:

● Neutral genetic variation: OA can negatively impact non-functional genetic variation in marine organisms (Lloyd et
al., 2016; Bitter et al., 2019). Indicators of these effects can be estimated from population genetic parameters such as the





number of alleles, heterozygosity, effective population size, inbreeding and population divergence (Schwartz et al., 2007). This assessment can be achieved using classic molecular markers (e.g., allozymes, microsatellites or mtDNA) or through high-throughput sequencing approaches (e.g., Lloyd et al., 2016; Bitter et al., 2019).

●    Functional genetic variation: Identifying and monitoring variation in ecologically important and heritable phenotypes traditionally involves the assessment of quantitative, functional genetic variation (Charmantier et al., 2014). For this, different approaches can be used, including quantitative trait locus (QTL) analysis, genome-wide association studies (GWAS), genome scan via restriction-site-associated DNA taqs (RAD-seq), and RNA sequencing (RNA-seq). Linkage and association mapping approaches (QTL and GWAS) allow the identification of regions of the genome that contribute to phenotypic variation and

divergence among populations (Savolainen et al., 2013). Similarly, RAD genome scans can affordably and efficiently identify: i) candidate loci underlying local adaptation; ii) signals of selection across the genomes and species range; iii) neutral and non-neutral genetic variation (Lowry et al., 2017). It is important to highlight here that functional genetic variation is not only related to structural differences in protein-coding loci but also in regulatory sequences that influence gene expression (Whitehead and Crawford, 2006). Therefore, approaches such as RNA-seq can help to identify candidate loci and regulatory

pathways involved in phenotypic responses to OA across spatio-temporal scales, as well as the mechanisms underlying phenotypic divergence, local adaptation and phenotypic plasticity in natural populations.

●    Mutation rates: Mutations are fundamental to evolution as they provide the ultimate source of novel and heritable variation on which selection can act. Thus, assessing mutation rates in natural populations allows us to infer ecological (e.g., population dynamics) and evolutionary (e.g., adaptation) responses of marine organisms to climate change factors such as OA

(Collins et al., 2020). In short-living organisms, drastic changes in environmental conditions can induce strong selection, driving rapid evolutionary change (sometimes as few as two generations) as a result of increases in the frequency of rare existing beneficial genetic variants or the generation of new beneficial mutations (Schaum et al., 2018). Estimates of mutation rates can be obtained from few loci (including genes of particular interest) or from whole-genomes using high-throughput sequencing technologies. In both cases, molecular variants can be screened among individuals and populations in order to

identify candidate genetic changes (de novo mutations and/or existing mutations) that might contribute to the ecological and evolutionary responses to particular environmental conditions such as OA (Schaum et al., 2018; Krasovec et al., 2020). This association between molecular variants such as single point mutations and the fitness of the organisms can be tested experimentally (e.g. Schaum et al., 2018; Waldvogel and Pfenninger, 2021), linking geno- and phenotypes in evolving populations. This linkage could be achieved using traditional physiological experiments or by measuring the effects of the

mutations on gene expression. Finally, mutations can be placed into a functional context by annotation of the protein sequences that host molecular variants using public databases such as the National Center for Biotechnology Information (NCBI) (e.g. Mock et al., 2017).





**4. Using "rates of change" to compare in-situ observations of biological impact and OA trends to allow regional and**
**global comparisons**

Monitoring any one of the parameters described above, over a period of time and in combination with coupled environmental data that includes at least two carbonate chemistry parameters, will allow the observer to better describe how their specific measured biological trait is changing in response to ocean acidification. In essence, by calculating, and then comparing, the rate at which the biological parameter is changing with the rate at which ocean acidification is occurring will identify how

closely correlated they are with each other. Such information will augment those experimental studies that identify mechanisms by which OA would act upon a biological process to generate a better conceptual understanding of how biological responses to OA will manifest themselves in the field. By the inclusion of other potentially important environmentally parameters, such hypoxia, temperature, pollution etc., it will also be possible to see how important OA is in driving biological response compared to the other environmental stressors for that particular biological trait. However, the overall purpose, and greater benefit, of

increasing the volume of OA biological monitoring is to collectively use all of these data together, in order to scale up site specific observations and to thereby understand OA impacts within and across species and populations, geographical locations and complex environments. For example, analyses could bring together multiple time-series datasets to understanding which specific elements of a trait are most affected, whether there is spatial / geographical variability in those responses observed and what are the key confounding drivers or processes that influence the biological response of organisms to OA. Below we

outline just one simple methodology for combining data gathered from different parameters and indices within a trait, in order to achieve that holistic overview of trait response. We appreciate that there could be other, more sophisticated data analysis techniques available, and undoubtably new ones will evolve in the future. For now, we present this simple method to illustrate a way in which those data that we are advocating should be collected, can be easily used to support greater understanding of OA's biological impacts.,

As the volume of information on the biological impact of OA has grown, there have been various efforts to synthesize this available knowledge and provide tools for meaningful comparisons among ecosystems and locations. For example, several approaches have been used to synthesize the literature on experimental biological studies including meta-analyses (e.g. Kroeker et al., 2013; Busch and McElhaney, 2016; Bednarsek et al., 2019) and semi-quantitative reviews (e.g. Wittmann and Pörtner, 2013). These approaches largely compare measurements of "effect size", where all observed responses are represented

as the relative change from a defined baseline or control condition specific to that observation (e.g. Kroeker et al., 2013). Whilst this approach overcomes the initial problem of data comparability, it still relies on assumptions that effect sizes are roughly representative of biological or ecological consequence. Whilst we acknowledge that further work will undoubtedly be needed to fully understand the ecological consequences associated with different scales of effect size and how this will change depending on the biological process being considered, the basic principle of using effect sizes to compare across different

indices remains useful.

At any given location, we propose that biological observations can be transformed into an effect size (e.g. in %) relative to a given reference time and modeling the evolution of this effect size through time allows to calculate a rate of biological change



as the total % change in the biological process divided by the time over which that change had happened. We propose that quantifying the relationship between these rates of biological change and the rates of chemical change associated with OA is

an effective way to assess the influence of that OA on the biological change observed. At any given location, this approach would thereby provide information on the risks and vulnerabilities associated with OA. A comparison of rates of change of specific chemical and biological indicators will also help in building a mechanistic understanding of the relationship between changes in relevant carbonate parameters and biological processes (e.g. Osborne et al., 2019). More importantly, it would allow comparison among overall biological responses across a wide range of locations.

As a general principle, in situations where OA is the primary driver of a biological response, the rate of that biological change is expected to be closely correlated to the rate of chemical change. The precise nature of this relationship (i.e. the direction and slope magnitude) will also depend on the sensitivity of the process, organism or ecosystem: at a given rate of chemical change, a sensitive organism is expected to show a stronger biological response than a less sensitive one. The rate of biological change is then the product of the interaction between biological sensitivity and rate of chemical change. By plotting together data from

various long-term studies, where each time series of biological and chemical observations is represented on a plot by a single point (defined by the rate of biological change on the x axis and the rate of chemical change on the y axis) it is possible to compare across different studies to define a generic relationship between OA and a specific biological trait (Fig. 2). It is important to remember that OA will not be the dominant driver at all locations. For example, a sensitive species exposed to low rate of chemical change may lead to lower rate of biological change than a more tolerant species exposed to high rate of

chemical change. Such deviations from the generic relationship would therefore identify locations where other biological drivers play a more important role or amplify (scenario 1 on Fig. 2) or mitigate the effect of OA that is being investigated (scenario 2 on Fig. 2). Furthermore, those deviations could highlight situations where the specific biological parameter measured was more or less sensitive to OA than others: for example, observations of calcification rate may all appear above the generic relationship line while observations of shell mass may all appear below the line. Both of these parameters,

calcification rate and shell mass, provide information on the calcification process but their relative positions on the plot would indicate differences in the mechanisms by which OA impacts upon those particular biological indices.

## 5. Points to consider when comparing the impacts of OA on biological traits derived from different observations and different places

To be meaningful and useful in the analysis proposed above, observed rates of biological change must be robust. If calculated

with too few data, these rates can severely under-estimate the real impacts. One of the key questions when evaluating an OA biological observation time-series is how long does that time-series need to be to ensure rates of change have been accurately quantified? The minimum duration of observation is largely dependent on both the specific sensitivity of the organism/ecosystem being observed and the underlying rate of chemical change. This concept is described theoretically in Fig. 3a which shows that, the longer the data are collected for, the more likely they are to provide an accurate estimate of the rate

of change over that time period. However, under higher rates of chemical changes, robust estimates of the rate of biological




change can be obtained over a shorter period of time. This illustrates the importance of backing up monitoring observations with targeted studies that can approximate the likely sensitivity of biological processes and parameters to OA stress. For example, generating a performance curve can provide an illustration of the effect of a given parameter (e.g. pH) on a biological response. Relationships between a stress and a biological response are often not linear and can be measured under laboratory

conditions for a given species and location (e.g. Ventura et al., 2016). These curves can then be used to provide estimates of the minimum duration of a biology monitoring program under different rates of chemical changes to properly quantify the rate of biological change.

Here we present an example using real data to illustrate the point. The performance curve between pH and larval abnormality was measured experimentally for the blue mussel *Mytilus edulis* population from the Gullmarsfjord in Sweden (Ventura et al.,

2016; medium sensitivity on Fig. 3b). The present average pH in surface seawater in the Gullmarsfjord is 8.1 (Dorey et al., 2013). Rates of pH change ranging from 0.001 (slow change corresponding to a decrease by 0.08 pH unit by 2100) to 0.006 pH unit decrease per year (fast change corresponding to a decrease by 0.48 pH unit by 2100) were compared. These rates were used to estimate the pH at any given time and the associated abnormality was calculated using the performance curve adapted from Ventura et al. (2016) (Fig. 4a). For each scenario of rate of chemical change, the rate of biological change was calculated

as the slope of the significant linear regression between time and abnormality (%) over the linear phase of the curve. For the blue mussel, there is a linear relationship between rates of chemical and biological change. In other words, for a given organism at a given location, the rate of biological change increases with the rate of chemical change (Fig. 4b).

It is possible then to use these data to estimate the number of years of observations to detect a negative impact. In the region where the mussels were collected, the current rate of chemical change in surface water is estimated as a decrease of 0.0044 pH

unit per year, corresponding to a decrease of 0.35 pH units by 2100 (Andersson et al., 2008). Based on our model, this would lead to an observed rate of biological change of 0.8861 % of abnormality per year by 2100 (Fig. 4). Using this model, it is possible to quantify the evolution of the rate of biological change over time (Fig. 5a) and then estimate how many years of observations are needed to calculate a good estimate. The rate of biological change increases with increasing years of observations (Fig. 5b). As a consequence, biological monitoring over a short period of time leads to an under-estimation of the

rate of biological change. For the mussel M. edulis in the Gullmarsfjord and under a realistic rate of chemical change, we can then estimate that at least 70 years of biological monitoring would be needed to calculate an accurate rate of biological change. It is worth noting that in the example presented above, the subject species was considered to be relatively insensitive to OA, resulting in a need for many decades of monitoring. For more sensitive biological subjects and processes, in areas of more rapid chemical change, the required length of observations will be much shorter. It is also important to note that other

parameters/stressors, short term variability, ecology and evolution can modulate the biological response and then influence the rate of biological change. As an example, we compared rates of biological changes in 3 'hypothetical model' organisms with different levels of sensitivity to pH (Fig. 3b). This scenario could represent different populations of the same species adapted to different levels of pH variability (e.g. Vargas et al., 2017, 2022) or evolution of the performance curve within one population





through time as a consequence of evolution. Assuming a rate of chemical change of 0.0044 pH unit per year (Andersson et al.,
2008), the rate of biological change observed by 2100 would be dependent on the species sensitivity (Fig. 6a).

For each level of sensitivity, it is possible to estimate the minimum number of years of observation needed to calculate an
accurate rate of biological change (with a 5% error; Fig. 6b). As expected, biological monitoring can be shorter for organisms
with higher sensitivity to pH. An accurate rate of biological change can be estimated after only 40 years for an organism with
high sensitivity as compared to 70 years for those with a medium sensitivity. For organisms with low sensitivity, the rate of
biological change is dramatically under-estimated after 80 years. As a consequence, if the purpose of a time-series is to
calculate a meaningful rate of biological change in as short a time as possible, it is recommended to focus on environments
with a high rate of chemical changes and biological indicators that are highly sensitive to OA. However, other sites and
processes of high ecological and socio-economic importance should not be neglected, even if it is likely to take many years
for a truly accurate relationship between biological response and OA to appear. It is essential to acknowledge that identifying
trends in regions with low rates of chemical change or in organisms with high tolerances to OA may require longer time but
that does not mean those impacts do not exist or are of low importance.

Estimation of species and population sensitivities are key parameters to consider in the design of a new biological program
aiming at documenting the biological impact of OA. Working with low sensitivity organisms would lead to a severe under-
estimation or unrealistically long data collection periods for an accurate estimation of the rate of biological change. As
sensitivity can be modulated by several parameters (e.g. evolution) within a population, it is recommended to regularly
calculate the rate of biological change and plot it over time. The rate of biological change is expected to increase with time
until saturation is reached (Fig. 5 & 6b). At saturation, the rate of biological change can then be considered as robust.

Finally, when comparing chemical and biological rates of change, it is also critical to ensure that the spatio-temporal resolution
of the carbonate chemistry data is directly relevant for the given species or ecosystem being observed. For example, chemistry
data collected in the surface water may not be relevant for a tidal organism experiencing much more variability than that
detected by the chemical observations. Similarly, it is important to remember that the carbonate chemistry experienced by a
given species is also dependent on its biology. For example, some species migrate over large distances and through very
different bodies of water or have different life-history stages living in different environments. Alternatively, some species,
mainly sessile, can alter temporal dynamics in the surrounding carbon chemistry through physiological processes such as
respiration, photosynthesis and calcification (e.g., sea grasses, kelps) (Pfister et al., 2019, Unsworth et al., 2012).

## 6. Conclusions

Assessing the vulnerabilities and adaptive capacity of marine ecosystems and coastal communities towards OA requires a
broad range of data. Measurements and calculations related to the SDG indicator 14.3.1 'Average marine acidity (pH)
measured at agreed suite of representative sampling stations' provide the chemical information required for such analyses.
However, achieving the SDG target 14.3 will also require correlating those observed chemical changes with the biological
response. This immense task can only be accomplished with a combination of approaches including laboratory and field





experimentation, modeling as well as biological observations. While the body of experimental evidence on OA impacts is growing rapidly, in-situ monitoring conducted specifically for identifying and quantifying biological responses to OA is far less common.

Our innovative approach goes beyond the simple monitoring of specified biological traits. We propose a strategy that aims to compare the relationships between rates of chemical and biological change under different circumstance, allowing the comparison across regional and global scales. We have shown that identifying robust rates of biological changes over a reasonable time (within a few decades) is dependent on both local species and ecosystem sensitivities and rate of chemical changes, and the interpretation of observed biological responses needs to be mindful of this. We show that it is possible to use

performance curves from experimental studies and data from chemical monitoring programs as a first order step to identify the sites for biological monitoring that will give the most rapid indication of OA impacts, as well as the minimum duration to measure robust rates of biological change. We also propose five fundamental ecosystem traits that optimize the probability of identifying OA impacts: 1) calcified organisms and calcification, 2) autotrophs and primary production, 3) heterotrophs and secondary production, 4) biodiversity and community structure, and 5) genetic adaptation. These traits span all marine

ecosystems, are anticipated to be sensitive to OA, and can be quantified using 'standardized' biological observations that in many cases are already collected alongside chemical observations. It needs to be highlighted that the specific choice of trait and respective parameters at different sites will depend on local practicalities, specific scientific questions and envisaged outcomes related to fitness of species and the health of marine populations and communities. The feasibility of obtaining these observations over the long term also must be considered, e.g. access to the marine ecosystem, technical and human scientific

capacity.

Selection of sites for biological observations should be based on answering a scientific question or to fit a specific stakeholder need. For example, sites can be selected for a practical reason (e.g. co-location with a long-term chemical observing site or at a marine station facilitating data collection), at a site of particular socio-economic or cultural value, or scientific interest (e.g. predicted low or high sensitivity to OA). We propose both a conceptual framework that would allow bridging chemical and

biological monitoring as well as the identifying the issues to consider when interpreting the collected data and evaluating the relevance of observed trends. We also argue that meaningful trends can be more rapidly identified at sites combining a high rate of chemical change with a high biological sensitivity to changes in the carbonate chemistry. However, this is only one of the many valid reasons to identify a monitoring site and as OA is only one of the many environmental parameters driving biological changes, other sites should not be neglected.

No single approach can explain the complexity of the biological consequences of OA. Monitoring is the ultimate tool to observe the impact of OA in the ocean but understanding and addressing the issue will require a combined approach with paleo investigations, field and laboratory experimentation, as well as modeling (Dupont et al., 2021). Each of these approaches is associated with its own set of strengths and limitations. Monitoring is realistic but often requires decades of data; experimentation can provide a mechanistic understanding over shorter periods of time but leads to practical limitations that

limits the realism (e.g. single stressors, lack of ecological relevance, short term; Riebesell and Gattuso, 2015). The real strength



comes from a combination of approaches. For example, we show that laboratory-based performance curves combined with data from observations of the carbonate system can be used to estimate the duration of a biological observing program to detect meaningful trends. Similarly, data from chemical and biological observations

could be combined to evaluate the performance curve of a given species in its natural environment (realized niche) and
compare with the curves from the laboratory (fundamental niche).

New data and information obtained when measuring biological indicators alongside chemical changes due to OA will allow the scientific community to contribute to the achievement of the vision of the UN Decade of Ocean Science for Sustainable Development 'The science we need for the ocean we want' (IOC-UNESCO, 2020), its outcomes and objectives. Biological observations assessing the impacts of OA will support several Ocean Decade challenges, in particular, Challenge 2
'Understand the effects of multiple stressors on ocean ecosystems, and develop solutions to monitor, protect, manage and restore ecosystems and their biodiversity under changing environmental, social and climate conditions.'. Furthermore, the implementation and related initiatives of this proposed strategy will be an integral part of delivering the Ocean Decade programme "Ocean Acidification Research for Sustainability (OARS) - Providing society with the observational and scientific evidence needed to sustainably identify, monitor, mitigate and adapt to ocean acidification; from local to global scales". The
strategies for biological observing and data analysis presented here offer a vision towards achieving OARS Outcome 4: Increase understanding of ocean acidification impacts to protect marine life by 2030. Specifically, by supporting the implementation of an established framework for biological observation, set within the existing ocean acidification monitoring framework, thereby providing the possibility to improve predictions of vulnerability and resilience to ocean acidification at all temporal and spatial scales. This paper presents a way to unifying biological field observations and so detect and compare
ocean acidification impacts across marine species and ecosystems, thus improving our understanding and knowledge about OA and thereby actively contribute to human well-being (Falkenberg et al., 2020).

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



## Genetic Adaptation

**Neutral genetic variation**

Classic molecular markers (e.g., allozymes, microsatellites or mtDNA); high-throughput sequencing approaches

**Mutation rates**

High-throughput sequencing technologies of few loci or whole genomes

**Functional genetic variation**

Quantitative trait locus (QTL) analysis; genome-wide association studies (GWAS); restriction-site-associated DNA tags (RAD-seq); RNA sequencing (RNA-seq)

| Calcifying Organisms and Calcification | Autotrophs and Primary Production | Heterotrophs and Secondary Production |
|---|---|---|
| **Relative prevalence and success of calcifying organisms** | **Biomass/standing stock** | **Biomass/standing stock** |
| Changes in biomass, abundance of biocalcifying species compared to non-calcifying species; inorganic to organic biomass ratios | Total chl a concentrations, phytoplankton cell abundance; microphyto-benthos biomass; biomass of macroalgae and seagrasses | Biomass per individual; numbers of individuals; average body sizes; percent cover, quantification of abundance and biomass of major functional or species groups |
| **Calcified biostructure morphology** | **Productivity** | **Productivity** |
| Weight, density, damage or abnormality, dissolution severity, or strength calcified biostructure | Carbon fixation rates, planktonic, macroalgal or seagrass growth rates | Gross estimates of pelagic and benthic secondary production from in-situ techniques to algorithms |
| **Rates of calcification** | **Phenology** | **Phenology** |
| Rates of calcification or dissolution | Timing of blooms or other rapid growth periods | Quantification of changes in the phenology of secondary producers |

## Biodiversity and Community Structure

**Taxonomic diversity and community composition**

Identification, quantification (number or biomass) of species, specific taxonomic or functional groups present within a community or assemblage at any given time

**Functional or trait diversity**

Identification, quantification (number or biomass) of functional, ecological, or behavioral traits

**Figure 1: Five fundamental ecosystem traits and their observable indicators.**





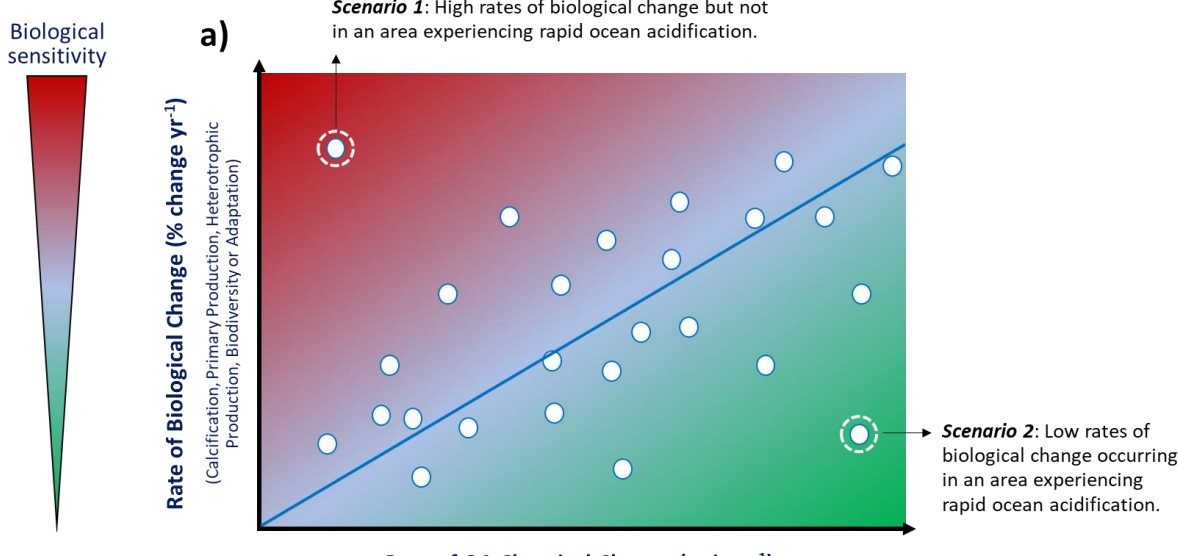

**Figure 2: Within each of the five proposed biological impact indicator categories (Calcification, Primary Production, Heterotrophic Production, Biodiversity, Adaptation) it is possible to combine the data from numerous individual time series stations in order to compare the rates of chemical change in with the rate of biological change across different geographical locations and between different biological contexts. Each individual data point on the graph represents**
**the summary information from a single time series that has coupled biological impact and carbonate chemistry observation data. With enough observations, it is possible to generate a generic relationship (blue line). Using the relative positions of data points on this plot it is possible to derive site specific information about the relative importance of OA in driving biological change at the individual time series stations. In scenario 1, the biological impact is largely being caused by other environmental drivers or the biological parameter that was measured is highly sensitive to OA.**
**In scenario 2, the measured biological impact parameter is not sensitive to OA or the biological impact is being mitigated by other environmental drivers or biological processes.**



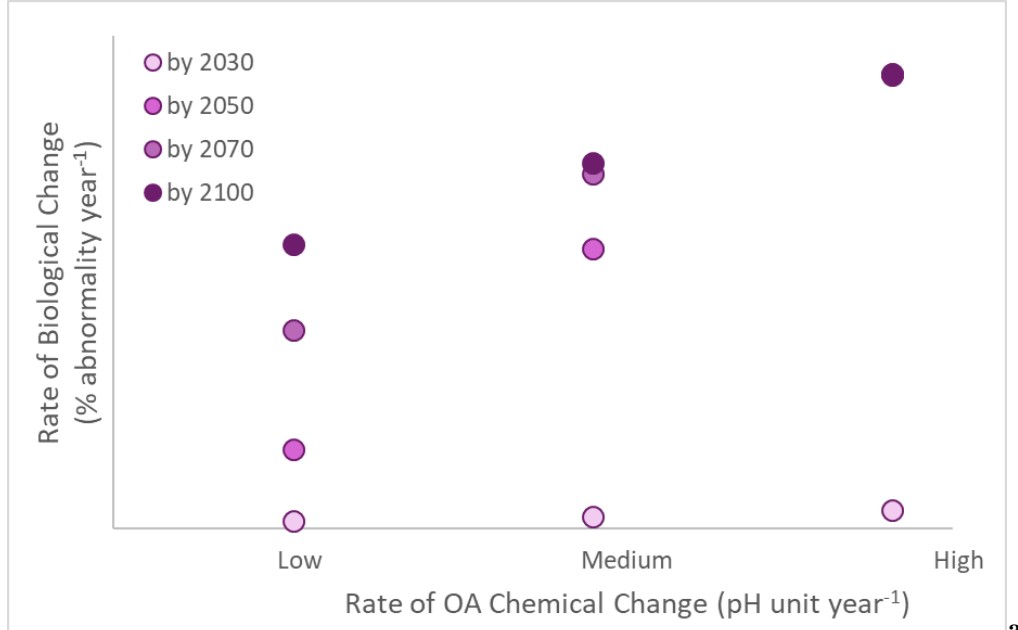

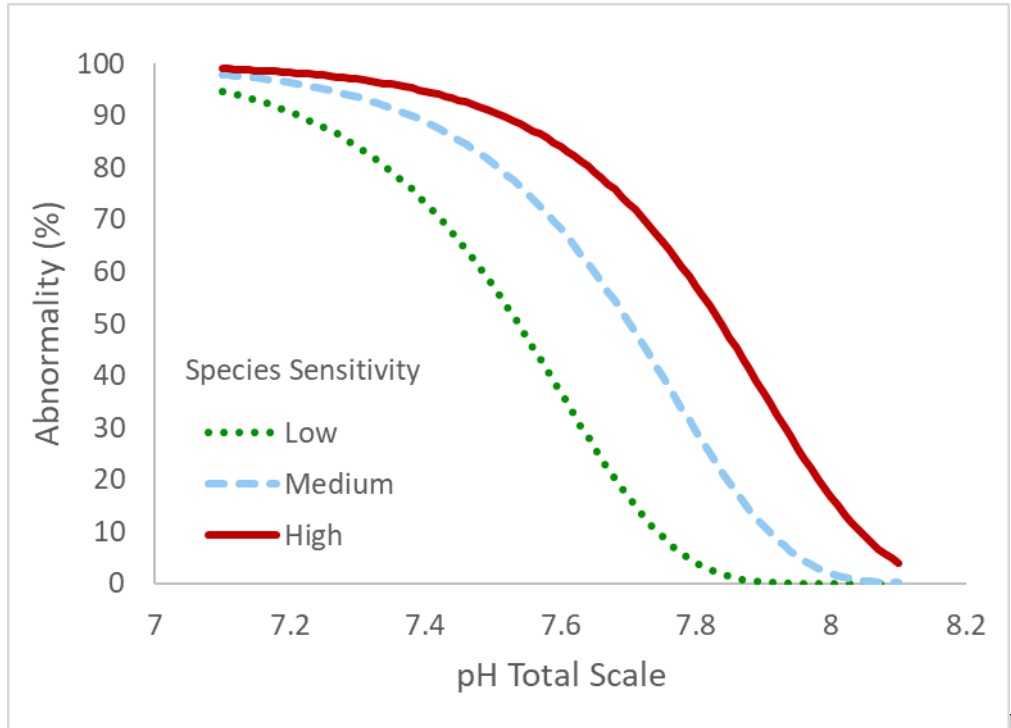

**Figure 3 – (a) Conceptual relationship between the rate of chemical change (e.g. pH unit per year) and the rate of biological change (e.g. % of change per year) under 3 different scenarios of rates of chemical change (low, medium and high). Rates of chemical and biological changes are positively correlated. The rates of biological change calculated by**




2100 are compared with rates calculated after 10, 30 and 50 years of observation. After 10 years, the rate of biological change is dramatically under-estimated as compared to the rate observed by 2100. Under low rate of chemical change, the rate of biological change is still under-estimated after 50 years of observation. Under high chemical rates of change, 780 similar biological rates of change can be calculated after 30, 50 and 80 years of observations. (b) Relationship between pH and larval abnormality (%) in 3 hypothetical organisms with different sensitivity: Low, Medium and High, with tipping points reached at pH 7.6, 7,75 and 7.9, respectively. The Medium scenario is based on the mussel Mytilus edulis (from Ventura et al., 2016).

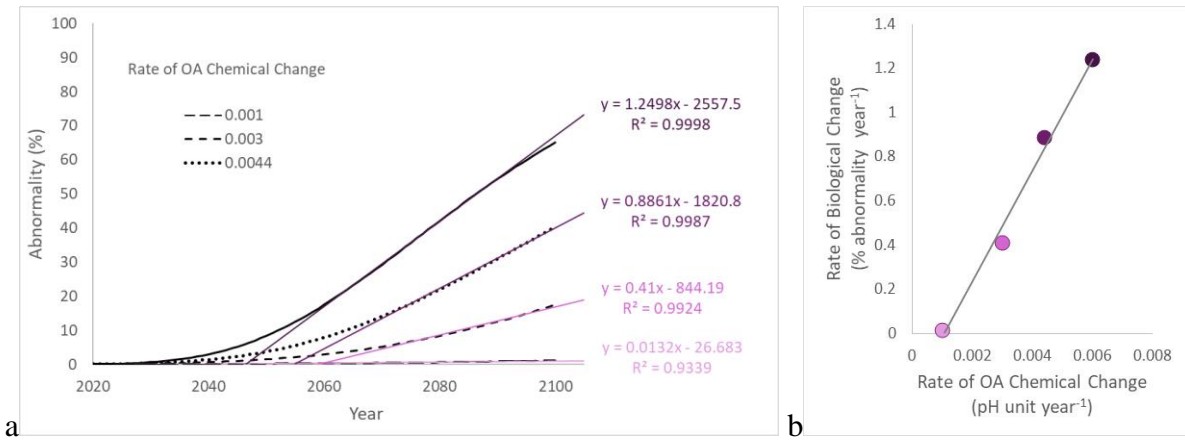

a        b

**Figure 4 – (a) Projection of larval abnormality under different scenarios of rate of chemical change. The rate of biological change was estimated for each scenario as the coefficient of the significant regression between time and abnormality (%) over the linear phase of the curve. (b) Relationship between the chemical and the biological rates of change.**

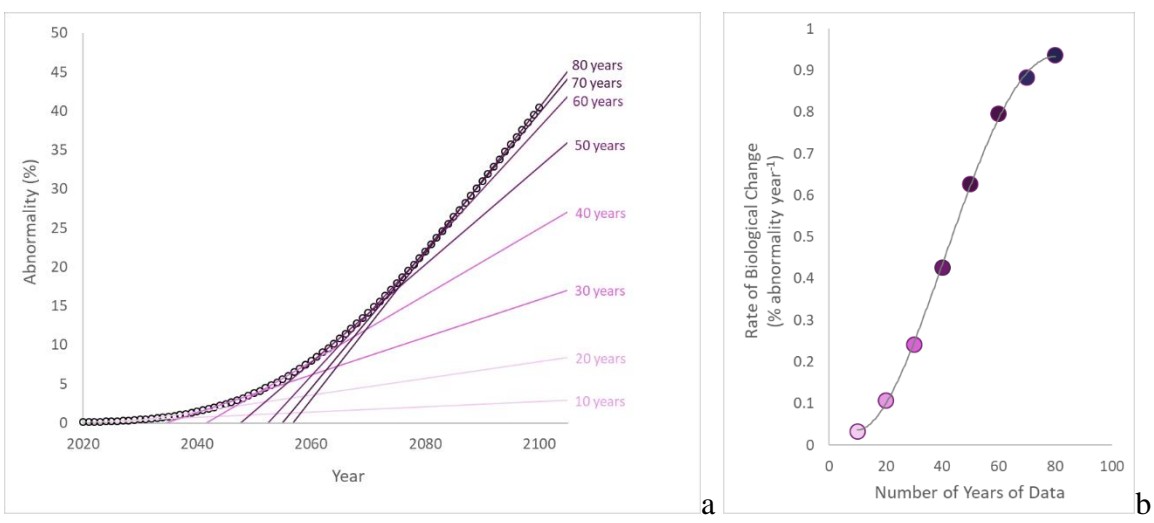

a      b





**Figure 5 – (a) Evaluation of the rate of biological change as the coefficient of the significant regression between time and abnormality over time (after collecting 10 years of data, 20 years of data, etc.). (b) Relationship between the number of years of data for the biological monitoring and the calculated rate of biological change.**

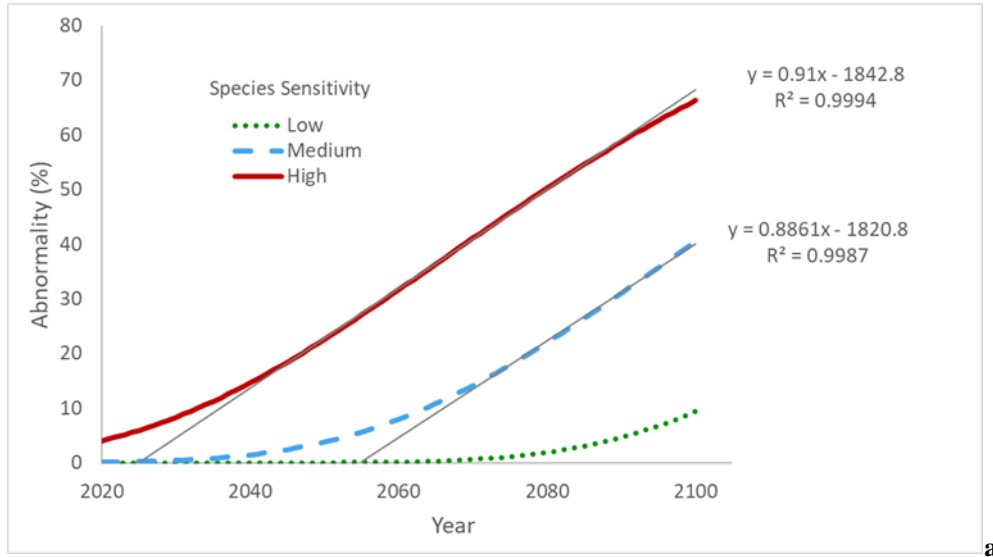

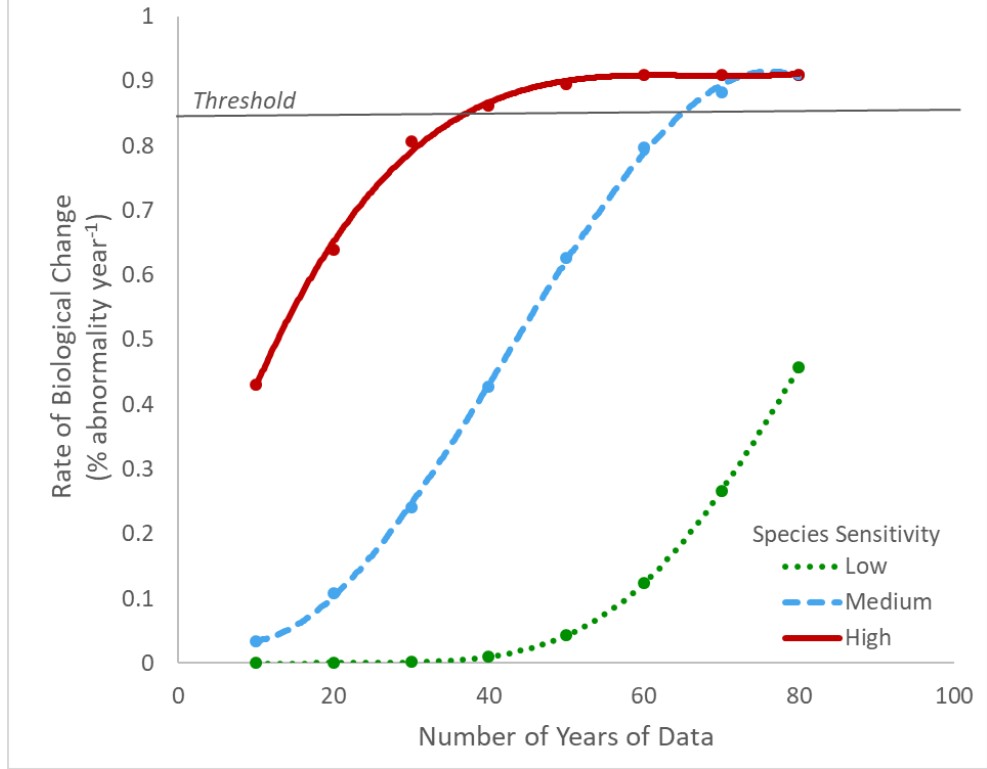



**Figure 6 – (a) Projection of larval abnormality under a rate of chemical change of 0.0044 pH unit per year (Andersson et al., 2008) for organisms with 3 different levels of sensitivity to pH (Low, Medium, High). The rate of biological change**
**was estimated for each scenario as the slope of the significant regression between time and abnormality (%) over the linear phase of the curve. For the species with low sensitivity, the linear phase was not reached, and it was not possible to calculate the rate of biological change. (b) Relationship between the number of years of data for the biological monitoring and the calculated rate of biological change for 3 organisms with different levels of sensitivity to pH (low, medium, high). It was considered that sufficient years of data were collected when the observed rate of biological change**
**was above the 0.86 threshold (5% from the maximum rate of change of 0.91 % of abnormality per year).**