# Peer review of "Unifying biological field observations to detect and compare ocean acidification impacts across marine species and ecosystems: What to monitor and why"

_EGUsphere, 2022_

## Author Response (AR1)

Dear Dr. Hoppema,

First of all, we would like to thank you for handling this manuscript.

We are also extremely grateful for the positive comments and suggestions of the two reviewers that are now addressed both on the discussion online and below, improving the quality of our manuscript.

Do not hesitate if you have any further comments or questions.

Best wishes,

Sam Dupont, on behalf of all co-authors.

REVIEWER #1

We would like to thank the reviewer for the positive comments and the thorough reading of the manuscript. We apologize for the typos that are now corrected in the next version of the manuscript. We have also added the suggested references and modify the Figure 1 to follow the same order as in the text. Some additional references have been added as requested. The next version of the manuscript takes into account all of the suggested changes.

-General comments:

This is an interesting manuscript proposing five broad classes of biological indicators that, when coupled with environmental observations, would allow the rate and severity of biological change in response to OA to be observed and compared. In addition, its approach allows the inclusion of a wide diversity of marine ecosystems in regional and global assessments.

The paper is well written, clear and concise. The manuscript is well documented although I miss some references in several sections (see individual comments below).

Therefore, I recommend this article for publication after minor revisions.

-Individual comments:

Line 92: "secondly"

Lines 119-122: provide references

Line 135: Remove "Observing" from the subtitle

Line 136: Figuerola et al., 2021?

Line 159: replace "Hofmann et al.; 2008" with "Hofmann et al. 2008;"

Line 164: Replace "Relative prevalence and success of calcifying organisms within an ecosystem" with "Biomass, abundance and percent cover"

Line 169: Replace "Calcified Biostructure Morphology" with "Skeletal morphology and composition"

Line 172: Add "porosity"

Line 198: Replace "Biomass/Standing Stock" with "Biomass and abundance"

Lines 202, 206, 210: provide references

Line 225: Replace "Biomass/Standing Stock" with "Biomass, abundance and percent cover"

Line 252: Add the following references: Kroeker et al., 2010, 2013; Hancock et al., 2020; Figuerola et al., 2021

Line 253: Add the following references after "…species loss": Hall-Spencer et al., 2008; Enochs et al., 2015

References:

Hall-Spencer, J. M. et al. Volcanic carbon dioxide vents show ecosystem effects of ocean acidification. Nature 454, 96–99 (2008).

Enochs, I. C. et al. Shift from coral to macroalgae dominance on a volcanically acidified reef. Nat. Clim. Chang. 5, 1–9 (2015).

Line 253: add the following references after "…trophic interactions": Kroeker et al., 2013b; Vizzini et al., 2017

References:

Kroeker, K. J., Gambi, M. C. & Micheli, F. Community dynamics and ecosystem simplification in a high-CO2 ocean. Proc. Natl. Acad. Sci. USA 110, 12721–12726 (2013b).

Vizzini, S. et al. Ocean acidification as a driver of community simplification via the collapse of higher-order and rise of lower-order consumers. Sci. Rep. 7, 1–10 (2017).

Line 256: references are not in chronological order along the text.

Lines 420 and 782: M. edulis in italics

Figure 1: Put the traits in the same order than mentioned in the text: first calcified organisms, second autotrophs…

Figure caption: Replace " Five fundamental ecosystem traits and their observable indicators." with "Five fundamental ecosystem traits identified as potentially sensitive to ocean acidification and their observable indicators."

REVIEWER #2

We would like to thank the reviewer for these valuable and constructive comments and suggestions. See below for our detailed responses.

**Reviewer comment**: It is timely that indicators are defined that help to attribute biological change to OA. This manuscript shows how experimental studies can inform monitoring strategies. The authors claim that there is a linear relationship between the rates of chemical and biological change. They claim that this concept could bring together a variety of indicators from different sites to attribute change to OA.

While the identified indicators can develop the EOV/EBV framework further, I find the correlation/linear regression of chemical and biological rates of change questionable.

**Author response**: We thank the reviewer for the comments and for highlighting an area of the manuscript where we haven't communicated the details of our approach as clearly as we should have. The underlying purpose of this paper is to introduce the concept that, if ocean acidification is the main (or strongly contributing) driver of change for a given ecosystem or biological parameter, it should be possible to describe a statistical relationship between the rate at which the carbonate chemistry is changing and the rate at which the observed biological parameter is changing. For simplicity, we chose to illustrate this point using a linear relationship (Figure 2) but we fully acknowledge that this relationship could follow one of many different models. Linear regression can be a good alternative and the null model used to test the data, so at least everyone involved in this approach will have the same starting point (the null model) although the ending point could be context-specific (depending on sensitivities, local changes, etc.) In fact, situations where the relationship model deviates from a simple linear one will allow us to explore the key modulating roles of other drivers and/or ecological interactions that may also be influencing the rate of biological change. We have now added text into the manuscript to clarify this point.

Added text: "While we use a linear relationship between the chemical and biological rates of change as an illustration in the Fig. 2, the precise nature of this relationship is currently unknown and may follow other pattern such as a non-linear tipping point responses (e.g. Wunderlink et al., 2021).

We also welcome the reviewer's reminder of the importance of developing indicators. such as those we propose, to develop the EOV/EBV framework further. We have added text to the introduction to better describe this and added a reference.

Added text: "If widely adopted, the approaches proposed in this paper would promote much greater co-location of biological and ocean acidification monitoring to help better understand the biological consequences of ocean acidification in the real world. This outcome aligns with previous recommendations (Muller-Karger et al., 2018b) which call for greater global connectivity between the collection of Essential Ocean Variables (EOVs) through the Global Ocean Observing System (GOOS), and Essential Biodiversity Variables (EBVs) from the Group on Earth Observations Biodiversity Observation Network (GEO BON). Although it should be noted that our inclusive approach encourages the collection of carbonate chemistry and biological data that goes beyond the defined set of observations that are recognised as either EOVs or EBVs. It should be further noted that several of the biological EOVs, remain to be further defined and the implemenaion of related observation alongside the physical and biogeochemical variables to be implemented (Miloslavich et al., 2018). Even so, our paper does support the ambitions outlined by Muller-Karger et al. (2018b) by proposing a practical way in which observing the biological impacts of ocean acidification can embrace the huge variety of biological observations that could be performed. Such an approach would not only facilitate the creation of completely new observing activities that include both biological and biogeochemical observations from the start, but also would provide guidance on how existing biochemical ocean acidification monitoring can add relevant biological observations to their suite of observations, and vice versa, to better appreciate the biological consequences of OA across regional and global scales."

**Reviewer comment**: Linear regressions, correlations and the comparison of dose-response curves (e.g. Fig. 4A) are not innovative methods. What is new is that they are applied in the context of OA to calculate monitoring durations in the field. There are a number of dose-response concepts in (eco)toxicology that can be instrumental to understanding and modelling OA effects (see e.g. species sensitivity distribution in Wittmann & Pörtner 2013). It would help the reader in understanding that using linear regression is a valid method to compare curves in Fig. 4A, if the authors would refer them to the respective literature in (eco)toxicology.

**Author response**: We fully agree with the reviewer that pH dose-responses curves are not expected to be linear (as for other environmental drivers). With enough data (e.g. long time series and/or high rates of chemical changes), it must be possible to use non-linear regressions to calculate the rate of biological change. However, in most cases the collected data from biological monitoring will not be sufficient to allow such calculations and, we would argue that, using a linear regression (following the approach presented in the Figure 4) can still be a useful tool. We have added text to section 5 in the manuscript to better describe our thinking around this.

Added text: "Ideally, the rate of biological change should be calculated using a non-linear model following the shape of the non-linear performance curve between pH and the tested parameter (e.g. Fig. 3b for % of abnormality). These performance curves can take many forms depending on the measured parameter and species. For example, common shapes of physiological performance curves include U-shape, sigmoidal, logarithmic, exponential, and inverted U-shape (Little &

Seebacher, 2021). However, in practice the shape of these curves is often unknown and it would not be possible to derive relevant rates of change from biological observations over relatively short period of time. Under these circumstances, using a linear regression is good alternative."

**Reviewer comment**: The model seems to be based on a single example, abnormality of Mytilus edulis. It is questionable that there is a linear relationship between chemical and biological change in all species and in all indicators. I suggest to use the large amount of laboratory results (OA-ICC at the database PANGAEA pangaea.de) and a variety of indicators to develop a more robust model.

**Author response**: The reviewer is quite correct, and the example projections made in this manuscript are based on a non-linear dose response (from Ventura et al. 2016). We then illustrate what the output would be using linear regression (and the associated errors). While we agree that using this approach with other species or even higher taxonomic groups (using synthesis and meta-analysis) is an extremely valuable thing to do we would see this as the natural next step, following on from this current manuscript. As such we consider this new body of work to be beyond the scope of this current manuscript that focusses primarily on presenting a methodology that can be applied by others to do exactly the kind of activity proposed by the reviewer.

**Reviewer comment**: The conclusion that insensitive systems require longer monitoring is trivial from the viewpoint of an experimental biologist as this must be considered when planning experiments and when interpreting results. While clearly illustrated in this manuscript, I doubt that this is new information to researchers involved in monitoring programs.

**Author response**: The reviewer makes a fair point about researchers already involved in biological monitoring, we would respectfully argue that this paper is not solely being directed towards that existing community. Coupled ocean acidification and biological impact observation/monitoring is largely absent in many countries and we feel it is useful to clarify even some of the most fundamental principles for other audiences, to promote a greater awareness and engagement in other stakeholders who we would hope to encourage to implement greater biological monitoring, complemented with carbonate chemistry measurements. Many of these stakeholders are non-biologist academics, environmental managers, funders of monitoring programmes and marine industry partners. We should also not forget that for a truly global picture to emerge we also need to make this manuscript accessible to those researchers looking to develop capacity and capability in countries with little or no experience in this type of monitoring. Consequently, we would prefer to keep this part of the conclusion, even if it may seem somewhat basic for some readers.

**Reviewer comment**: An innovative approach would allow to consider processes such as transgenerational effects of OA (see e.g. Parker et al. 2015) and multiple drivers of biological change. I suggest to include a figure that at least conceptually depicts how dose-response curves may be affected.

**Author response**: The reviewer makes a good point. We did think that we had implicitly covered the potential modulating role of transgenerational acclimation, evolution or other drivers in leading to change in species sensitivity this in Figure 6. To make this point more clearly we have modified the text.

Modified text: "It is also important to note that other parameters/stressors, short term variability, ecology and evolution can modulate the biological response and then influence the rate of biological change. As an example, we compared rates of biological changes in 3 'hypothetical model' organisms with different levels of sensitivity to pH (Fig. 3b). This scenario could represent different populations of the same species adapted to different levels of pH variability (e.g. Vargas et al., 2017, 2022) or with different modulating factors (e.g. temperature or food concentration) or evolution of the performance curve within one population through time as a consequence of transgenerational adaptation or evolution (Parker et al., 2015; Thor & Dupont, 2015; De Wit et al., 2016)."

**Reviewer comment**: Specific comments

L 63-66: This sentence suggests that there are few field (monitoring?) studies on OA effects outside areas of unusually high CO2 levels. If this is so, I suggest to cite them and briefly summarize the state of the art here or in the following paragraph. This would give the reader a brief overview of what could be intensified/improved in the future. Otherwise, consider rephrasing this sentence and the sentence in l 462-464 of the conclusions. Also good to know here: Are there any monitoring data that allow the analysis of biological OA effects, and this just has not been done yet?

**Author response**: As far as we know, linking chemical and biological monitoring data has not been done yet. We hope that this manuscript will stimulate such exercise after identification of suitable datasets but also motivate co-localized monitoring in stations that have already either chemical or biological observations.

In the sentence 63-66, we referred to discrete observations and experimentation in the field. An example (Hall-Spencer et al. 2008) has been added

**Reviewer comment**: L 75 ff Introduction: More explicitly name and address EOVs, EBVs, MBON here and in the conclusions. Clearly point out and explain the contribution of ecophysiology/experimental studies to improving the EOV, EBV framework with respect to OA here and in the conclusions (e.g. Hayes et al. 2015, Kissling et al. 2018, Pereira et al. 2013). I.e. what are the gaps this paper fills?

**Author response:** The benefits of implementing our proposed response to the EOV-EBV framework have now been acknowledged in the introduction and conclusions.

**Reviewer comment**: L 132: Replace "or" with "and". Physiological data are important to attribute impacts to any environmental driver. Ecological data alone do not allow this. And vice versa.

**Author response:** Done

**Reviewer comment**: L 137 ff: State how these ecosystem traits and indicators relate to EOVs and EBVs (see references above).

**Author response:** Done

**Reviewer comment**: L 139 ff: In reality, it most likely "will depend on questions or concerns of the investigator". However, in monitoring, the choice of indicators should be consistent across sites and not depend on "questions or concerns of the investigator" locally. Otherwise, comparison is hampered. Investigators across sites should at least try to agree on common indicators and monitoring strategies. Consider including this point in line 339 ff, section 4, and the conclusions.

**Author response:** We would argue is aimed at solving exactly this issue. By adapting a trait-based approach we provide the community to both contribute to a broad scale comparison of biological impacts, whilst also allowing them to observe those biological parameters that have the most local importance to them.

**Reviewer comment**: L 144-146: Provide reference.

**Author response:** Done

**Reviewer comment**: L 250-251: Insert citation Wittman & Pörtner 2013.

**Author response:** Done

**Reviewer comment**: L 410: Why was linear regression used? Drawn from toxicological concepts? If so, please cite respective literature.

**Author response** : The dose response curve used for this exercise is not linear (Ventura et al. 2016) but the calculations assume an incomplete dataset (as it would be over the first decades of a new biological monitoring program). It is then not possible to use a more relevant modeling. Using a linear regression over the linear part of the curve is then a simple and efficient tool. See above for addition in the text.

**Reviewer comment**: L 411-412: Please provide evidence that supports this general statement.

**Author response:** This sentence was rephrased to clarify that it is what is observed in our projections.

**Reviewer comment**: L 785: Figure 4: Add pH scales to the x-axis to clarify that these are dose-response curves.

**Author response:** There is not pH on the x-axis for the Figure 4. If it is a reference to the Figure 3, it is a rate (pH unit per time) and the scale is not so relevant for that exercise.

**Reviewer comment**: Technical corrections

**Author response:** All done, sorry for the typos and we thank the reviewer for bringing these to our attention through their thorough reading of the manuscript.

L 51: Delete "process".

L 63-66: Long sentence, consider revising.

L 66: Provide full citation for OA-ICC.

L 73: Sentence "Biological observations…": Replace "it" with "they".

L 141: Add "should be" in front of "associated".

L193: Delete "might".

L 326: Insert "GenBank and the European Nucleotide Archive (ENA)" after "(NCBI)".

L 337: Replace "environmentally" with "environmental"; add "as" after "such".

L 340: Delete comma.

References

Hayes et al. 2015 Identifying indicators and essential variables for marine ecosystems https://doi.org/10.1016/j.ecolind.2015.05.006

Kissling, W.D., Walls, R., Bowser, A. et al. Towards global data products of Essential Biodiversity Variables on species traits. Nat Ecol Evol 2, 1531–1540 (2018). https://doi.org/10.1038/s41559-018-0667-3

Parker LM, O'Connor WA, Raftos DA, Pörtner HO, Ross PM. Persistence of Positive Carryover Effects in the Oyster, Saccostrea glomerata, following Transgenerational Exposure to Ocean Acidification. PLoS One. 2015 Jul 6;10(7):e0132276. doi: 10.1371/journal.pone.0132276.

Pereira et al. 2013 Essential Biodiversity Variables - A global system of harmonized observations is needed to inform scientists and policy-makers DOI: 10.1126/science.1229931